

# Detecting periodicities with Gaussian processes

Nicolas Durrande[1], James Hensman[2], Magnus Rattray[3] and Neil D. Lawrence[4]

[1] Institut Fayol—LIMOS, Mines Saint-Étienne, Saint-Étienne, France
[2] CHICAS, Faculty of Health and Medicine, Lancaster University, Lancaster, United Kingdom
[3] Faculty of Life Sciences, University of Manchester, Manchester, United Kingdom
[4] Department of Computer Science and Sheffield Institute for Translational Neuroscience, University of Sheffield, Sheffield, United Kingdom

## ABSTRACT

We consider the problem of detecting and quantifying the periodic component of a function given noise-corrupted observations of a limited number of input/output tuples. Our approach is based on Gaussian process regression, which provides a flexible non-parametric framework for modelling periodic data. We introduce a novel decomposition of the covariance function as the sum of periodic and aperiodic kernels. This decomposition allows for the creation of sub-models which capture the periodic nature of the signal and its complement. To quantify the periodicity of the signal, we derive a periodicity ratio which reflects the uncertainty in the fitted sub-models. Although the method can be applied to many kernels, we give a special emphasis to the Matérn family, from the expression of the reproducing kernel Hilbert space inner product to the implementation of the associated periodic kernels in a Gaussian process toolkit. The proposed method is illustrated by considering the detection of periodically expressed genes in the *arabidopsis* genome.

## INTRODUCTION

The periodic behaviour of natural phenomena arises at many scales, from the small wavelength of electromagnetic radiations to the movements of planets. The mathematical study of natural cycles can be traced back to the nineteenth century with Thompson's harmonic analysis for predicting tides (*Thomson, 1878*) and Schuster's investigations on the periodicity of sunspots (*Schuster, 1898*). Amongst the methods that have been considered for detecting and extracting the periodic trend, one can cite harmonic analysis (*Hartley, 1949*), folding methods (*Stellingwerf, 1978*; *Leahy et al., 1983*) which are mostly used in astrophysics and periodic autoregressive models (*Troutman, 1979*; *Vecchia, 1985*). In this article, we focus on the application of harmonic analysis in reproducing kernel Hilbert spaces (RKHS) and on the consequences for Gaussian process modelling. Our approach provides a flexible framework for inferring both the periodic *and* aperiodic components of sparsely sampled and noise-corrupted data, providing a principled means for quantifying the degree of periodicity. We demonstrate our proposed method on the problem of identifying periodic genes in gene expression time course data, comparing performance with a popular alternative approach to this problem.

Corresponding author
Nicolas Durrande, durrande@emse.fr

Harmonic analysis is based on the projection of a function onto a basis of periodic functions. For example, a natural method for extracting the $2\pi$-periodic trend of a function $f$ is to decompose it in a Fourier series:

$$f(x) \rightarrow f_p(x) = a_1 \sin(x) + a_2 \cos(x) + a_3 \sin(2x) + a_4 \cos(2x) + \cdots \tag{1}$$

where the coefficients $a_i$ are given, up to a normalising constant, by the $L^2$ inner product between $f$ and the elements of the basis. However, the phenomenon under study is often observed at a limited number of points, which means that the value of $f(x)$ is not known for all $x$ but only for a small set of inputs $\{x_1, \ldots, x_n\}$ called the observation points. With this limited knowledge of $f$, it is not possible to compute the integrals of the $L^2$ inner product so the coefficients $a_i$ cannot be obtained directly. The observations may also be corrupted by noise, further complicating the problem.

A popular approach to overcome the fact that $f$ is partially known is to build a mathematical model $m$ to approximate it. A good model $m$ has to take into account as much information as possible about $f$. In the case of noise-free observations it interpolates $f$ for the set of observation points $m(x_i) = f(x_i)$ and its differentiability corresponds to the assumptions one can have about the regularity of $f$. The main body of literature tackling the issue of interpolating spatial data is scattered over three fields: (geo-)statistics (*Matheron, 1963*; *Stein, 1999*), functional analysis (*Aronszajn, 1950*; *Berlinet & Thomas-Agnan, 2004*) and machine learning (*Rasmussen & Williams, 2006*). In the statistics and machine learning framework, the solution of the interpolation problem corresponds to the expectation of a Gaussian process, $Z$, which is conditioned on the observations. In functional analysis, the problem reduces to finding the interpolator with minimal norm in a RKHS $\mathcal{H}$. As many authors pointed out (for example *Berlinet & Thomas-Agnan (2004)* and *Scheuerer, Schaback & Schlather (2011)*), the two approaches are closely related. Both $Z$ and $\mathcal{H}$ are based on a common object which is a positive definite function of two variables $k(.,.)$. In statistics, $k$ corresponds to the covariance of $Z$ and for the functional counterpart, $k$ is the reproducing kernel of $\mathcal{H}$. From the regularization point of view, the two approaches are equivalent since they lead to the same model $m$ (*Wahba, 1990*). Although we will focus hereafter on the RKHS framework to design periodic kernels, we will also take advantage of the powerful probabilistic interpretation offered by Gaussian processes.

We propose in this article to build the Fourier series using the RKHS inner product instead of the $L^2$ one. To do so, we extract the sub-RKHS $\mathcal{H}_p$ of periodic functions in $\mathcal{H}$ and model the periodic part of $f$ by its orthogonal projection onto $\mathcal{H}_p$. One major asset of this approach is to give a rigorous definition of non-periodic (or aperiodic) functions as the elements of the sub-RKHS $\mathcal{H}_a = \mathcal{H}_p^{\perp}$. The decomposition $\mathcal{H} = \mathcal{H}_p \oplus \mathcal{H}_a$ then allows discrimination of the periodic component of the signal from the aperiodic one. Although some expressions of kernels leading to RKHS of periodic functions can be found in the literature (*Rasmussen & Williams, 2006*), they do not allow to extract the periodic part of the signal. Indeed, usual periodic kernels do not come with the expression of an aperiodic kernel. It is thus not possible to obtain a proper decomposition of the space as the direct sum of periodic and aperiodic subspaces and the periodic sub-model cannot be rigorously obtained.

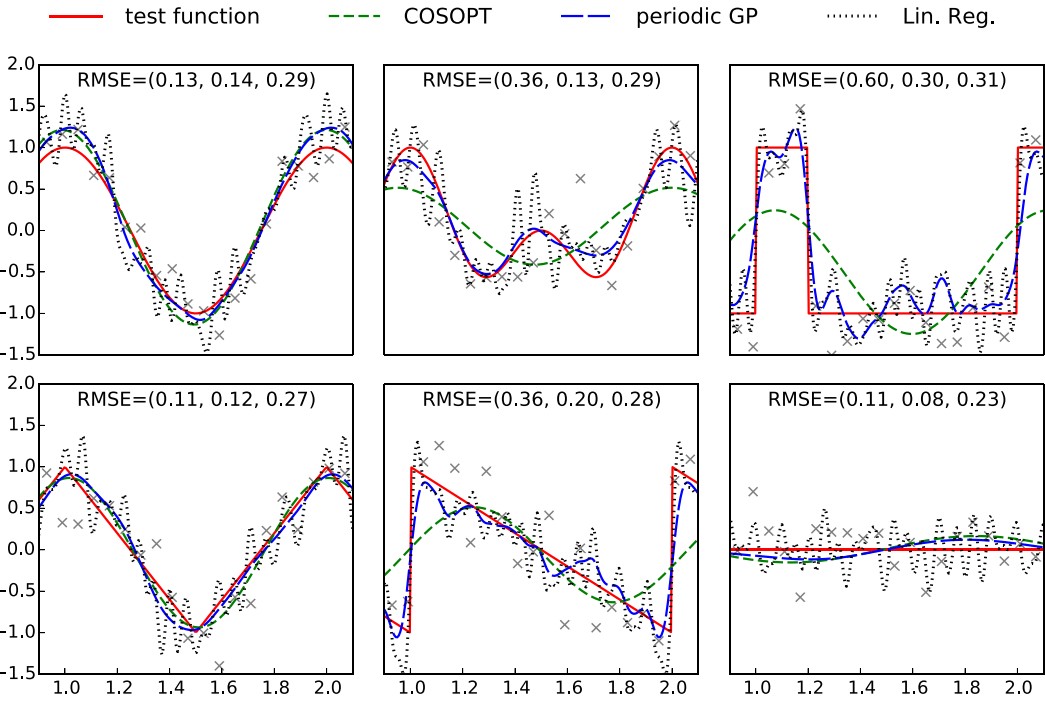

**Figure 1** **Plots of the benchmark test functions, observation points and fitted models.** For an improved visibility, the plotting region is limited to one period. The RMSE is computed using a grid of 500 evenly spaced points spanning $[0, 3]$, and the values indicated on each subplot correspond respectively to COSOPT, the periodic Gaussian process model and linear regression. The Python code used to generate this figure is provided as Jupyter notebook in Supplemental Information 3.

The last part of this introduction is dedicated to a motivating example. In 'Kernels of Periodic and Aperiodic Subspaces,' we focus on the construction of periodic and aperiodic kernels and on the associated model decomposition. 'Application to Matérn Kernels' details how to perform the required computations for kernels from the Matérn familly. 'Quantifying the Periodicity t' introduces a new criterion for measuring the periodicity of the signal. Finally, the last section illustrates the proposed approach on a biological case study where we detect, amongst the entire genome, the genes showing a cyclic expression.

The examples and the results presented in this article have been generated with the version 0.8 of the python Gaussian process toolbox *GPy*. This toolbox, in which we have implemented the periodic kernels discussed here, can be downloaded at http://github.com/SheffieldML/GPy. Furthermore, the code generating the Figs. 1–3 is provided in the Supplemental Information 3 as Jupyter notebooks.

## Motivating example

To illustrate the challenges of determining a periodic function, we first consider a benchmark of six one dimensional periodic test functions (see Fig. 1 and Appendix A). These functions include a broad variety of shapes so that we can understand the effect of shape on methods with different modelling assumptions. A set $X = (x_1, \ldots, x_{50})$ of equally spaced observation points is used as training set and a $\mathcal{N}(0, 0.1)$ observation noise is added to each evaluation of the test function: $F_i = f(x_i) + \varepsilon_i$ (or $F = f(X) + \varepsilon$ with vector

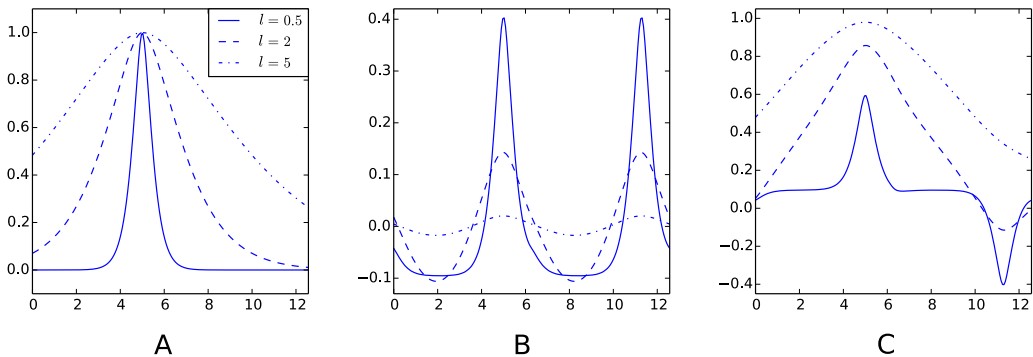

**Figure 2** **Examples of decompositions of a kernel as a sum of a periodic and aperiodic sub-kernels.** (A) Matérn 3/2 kernel $k(.,5)$. (B) Periodic sub-kernel $k_p(.,5)$. (C) Aperiodic sub-kernel $k_a(.,5)$. For these plots, one of the kernels variables is fixed to 5. The three graphs on each plot correspond to a different value of the lengthscale parameter $\ell$. The input space is $D = [0, 4\pi]$ and the cut-off frequency is $q = 20$. The Python code used to generate this figure is provided as Jupyter notebook in Supplemental Information 3.

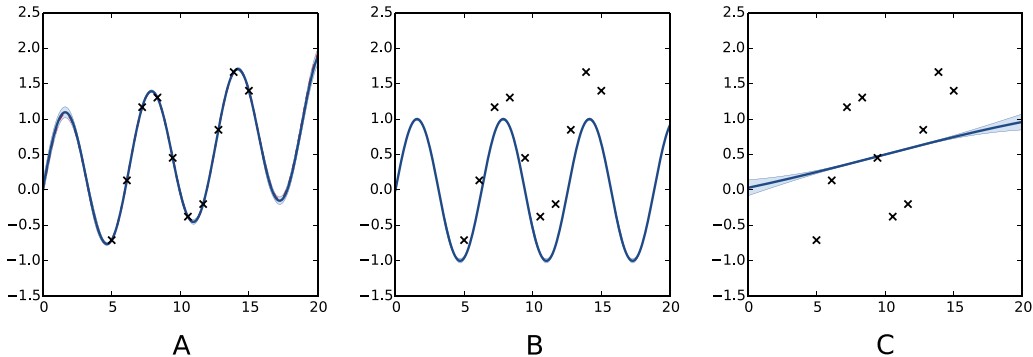

**Figure 3** **Decomposition of a Gaussian process fit.** (A) full model $m$; (B) periodic portion $m_p$ and (C) aperiodic portion $m_a$. Our decomposition allows for recognition of both periodic and aperiodic parts. In this case maximum likelihood estimation was used to determine the parameters of the kernel, we recovered $(\sigma_p^2, \ell_p, \sigma_a^2, \ell_a) = (52.96, 5.99, 1.18, 47.79)$. The Python code used to generate this figure is provided as Jupyter notebook in Supplemental Information 3.

notations). We consider three different modelling approaches to compare the facets of different approaches based on harmonic analysis:

- COSOPT (*Straume, 2004*), which fits cosine basis functions to the data,
- Linear regression in the weights of a truncated Fourier expansion,
- Gaussian process regression with a periodic kernel.

**COSOPT.** COSOPT is a method that is commonly used in biostatistics for detecting periodically expressed genes (*Hughes et al., 2009*; *Amaral & Johnston, 2012*). It assumes the following model for the signal:

$$y(x) = \alpha + \beta \cos(\omega x + \varphi) + \varepsilon, \tag{2}$$

where $\varepsilon$ corresponds to white noise. The parameters $\alpha$, $\beta$, $\omega$ and $\varphi$ are fitted by minimizing the mean square error.

**Linear regression.** We fit a more general model with a basis of sines and cosines with periods $1, 1/2 \ldots, 1/20$ to account for periodic signal that does not correspond to a pure sinusoidal signal.

$$y(x) = \alpha + \sum_{i=1}^{20} \beta_i \cos(2\pi i x) + \sum_{i=1}^{20} \gamma_i \sin(2\pi i x) + \varepsilon. \tag{3}$$

Again, model parameters are fitted by minimizing the mean square error which corresponds to linear regression over the basis weights.

**Gaussian Process with periodic covariance function.** We fit a Gaussian process model with an underlying periodic kernel. We consider a model,

$$y(x) = \alpha + y_p(x) + \varepsilon, \tag{4}$$

where $y_p$ is a Gaussian process and where $\alpha$ should be interpreted as a Gaussian random variable with zero mean and variance $\sigma_\alpha^2$. The periodicity of the phenomenon is taken into account by choosing a process $y_p$ such that the samples are periodic functions. This can be achieved with a kernel such as

$$k_p(x, x') = \sigma^2 \exp\left( -\frac{\sin^2\left(\omega(x - x')\right)}{\ell} \right) \tag{5}$$

or with the kernels discussed later in the article. For this example we choose the periodic Matérn 3/2 kernel which is represented in Fig. 2B. For any kernel choice, the Gaussian process regression model can be summarized by the mean and variance of the conditional distribution:

$$
\begin{aligned}
m(x) &= \mathrm{E}[y(x)|y(X) = F] = k(x, X)(k(X, X) + \tau^2 I)^{-1} F \\
v(x) &= \mathrm{Var}[y(x)|y(X) = F] = k(x, x) - k(x, X)(k(X, X) + \tau^2 I)^{-1} k(X, x)
\end{aligned}
\tag{6}
$$

where $k = \sigma_\alpha^2 + k_p$ and $I$ is the $50 \times 50$ identity matrix. In this expression, we introduced matrix notation for $k$: if $A$ and $B$ are vectors of length $n$ and $m$, then $k(A, B)$ is a $n \times m$ matrix with entries $k(A, B)_{i,j} = k(A_i, B_j)$. The parameters of the model $(\sigma_\alpha^2, \sigma^2, \ell, \tau^2)$ can be obtained by maximum likelihood estimation.

The models fitted with COSOPT, linear regression and the periodic Gaussian process model are compared in Fig. 1. It can be seen that the latter clearly outperforms the other models since it can approximate non sinusoidal patterns (in opposition to COSOPT) while offering a good noise filtering (no high frequencies oscillations corresponding to noise overfitting such as for linear regression).

The Gaussian process model gives an effective non-parametric fit to the different functions. In terms of root mean square error (RMSE) in each case, it is either the best performing method, or it performs nearly as well as the best performing method. Both linear regression and COSOPT can fail catastrophically on one or more of these examples.

Although highly effective for purely periodic data, the use of a periodic Gaussian processes is less appropriate for identifying the periodic component of a pseudo-periodic function such as $f(x) = \cos(x) + 0.1 \exp(-x)$. An alternative suggestion is

Durrande et al. (2016), *PeerJ Comput. Sci.*, DOI 10.7717/peerj-cs.50

to consider a pseudo-periodic Gaussian process $y = y_1 + y_p$ with a kernel given by the sum of a usual kernel $k_1$ and a periodic one $k_p$ (see e.g., *Rasmussen & Williams, (2006)*). Such a construction allows decomposition of the model into a sum of sub-models $m(x) = \mathrm{E}[y_1(x)|y(X) = F] + \mathrm{E}[y_p(x)|y(X) = F]$ where the latter is periodic (see 'Decomposition in periodic and aperiodic sub-models' for more details). However, the periodic part of the signal is scattered over the two sub-models so it is not fully represented by the periodic sub-model. It would therefore be desirable to introduce new covariance structures that allow an appropriate decomposition in periodic and non-periodic sub-models in order to tackle periodicity estimation for pseudo-periodic signals.

# KERNELS OF PERIODIC AND APERIODIC SUBSPACES

The challenge of creating a pair of kernels that stand respectively for the periodic and aperiodic components of the signal can be tackled using the RKHS framework. We detail in this section how decomposing a RKHS into a subspace of periodic functions and its orthogonal complement leads to periodic and aperiodic sub-kernels.

## Fourier basis in RKHS

We assume in this section that the space $\mathcal{H}_p$ spanned by a truncated Fourier basis

$$B(x) = \left( \sin\left(\frac{2\pi}{\lambda}x\right), \ldots, \cos\left(\frac{2\pi}{\lambda}qx\right) \right)^{\top} \qquad (7)$$

is a subspace of the RKHS $\mathcal{H}$. Under this hypothesis, it is straightforward to confirm that the reproducing kernel of $\mathcal{H}_p$ is

$$k_p(x, x') = B^{\top}(x)G^{-1}B(x') \qquad (8)$$

where $G$ is the Gram matrix of $B$ in $\mathcal{H}$: $G_{i,j} = \langle B_i, B_j \rangle_{\mathcal{H}}$. Hereafter, we will refer to $k_p$ as the *periodic kernel*. In practice, the computation of $k_p$ requires computation of the inner product between sine and cosine functions in $\mathcal{H}$. We will see in the next section that these computations can be done analytically for Matérn kernels. For other kernels, a more comprehensive list of RKHS inner products can be found in *Berlinet & Thomas-Agnan* (*2004*, Chap. 7).

The orthogonal complement of $\mathcal{H}_p$ in $\mathcal{H}$ can be interpreted as a subspace $\mathcal{H}_a$ of *aperiodic* functions. By construction, its kernel is $k_a = k - k_p$ (*Berlinet & Thomas-Agnan, 2004*). An illustration of the decomposition of Matérn 3/2 kernels is given in Fig. 2. The decomposition of the kernel comes with a decomposition of the associated Gaussian process in to two independent processes and the overall decompositions can be summarised as follow:

$$\mathcal{H} = \mathcal{H}_p \overset{\perp}{+} \mathcal{H}_a \leftrightarrow k = k_p + k_a \leftrightarrow y = y_p \overset{\perp\!\!\!\perp}{+} y_a. \qquad (9)$$

Many stationary covariance functions depend on two parameters: a variance parameter $\sigma^2$, which represents the vertical scale of the process and a lengthscale parameter, $\ell$, which represents the horizontal scale of the process. The sub-kernels $k_a$ and $k_p$ inherit these parameters (through the Gram matrix $G$ for the latter). However, the decomposition $k = k_p + k_a$ allows us to set the values of those parameters separately for each sub-kernel

in order to increase the flexibility of the model. The new set of parameters of $k$ is then $(\sigma_p^2, \ell_p, \sigma_a^2, \ell_a)$ with an extra parameter $\lambda$ if the period is not known.

Such reparametrisations of $k_p$ and $k_a$ induce changes in the norms of $\mathcal{H}_p$ and $\mathcal{H}_a$. However, if the values of the parameters are not equal to zero or $+\infty$, these spaces still consist of the same elements so $\mathcal{H}_p \cap \mathcal{H}_a = \varnothing$. This implies that the RKHS generated by $k_p + k_a$ corresponds to $\mathcal{H}_p + \mathcal{H}_a$ where the latter are still orthogonal but endowed with a different norm. Nevertheless, the approach is philosophically different since we build $\mathcal{H}$ by adding two spaces orthogonally whereas in Eq. (9) we decompose an existing space $\mathcal{H}$ into orthogonal subspaces.

### Decomposition in periodic and aperiodic sub-models

The expression $y = y_p + y_a$ of Eq. (9) allows to introduce two sub-models corresponding to conditional distributions: a periodic one $y_p(x)|y(X) = F$ and an aperiodic one $y_a(x)|y(X) = F$. These two distributions are Gaussian and their mean and variance are given by the usual Gaussian process conditioning formulas

$$
\begin{aligned}
m_p(x) &= \mathrm{E}[y_p(x)|y(X) = F] = k_p(x,X)k(X,X)^{-1}F \\
m_a(x) &= \mathrm{E}[y_a(x)|y(X) = F] = k_a(x,X)k(X,X)^{-1}F,
\end{aligned}
\tag{10}
$$

$$
\begin{aligned}
v_p(x) &= \mathrm{Var}[y_p(x)|y(X) = F] = k_p(x,x) - k_p(x,X)k(X,X)^{-1}k_p(X,x) \\
v_a(x) &= \mathrm{Var}[y_a(x)|y(X) = F] = k_a(x,x) - k_a(x,X)k(X,X)^{-1}k_a(X,x).
\end{aligned}
\tag{11}
$$

The linearity of the expectation ensures that the sum of the sub-models means is equal to the full model mean:

$$
\begin{aligned}
m(x) &= \mathrm{E}[y_p(x) + y_a(x)|y(X) = F] = \mathrm{E}[y_p(x)|y(X) = F] + \mathrm{E}[y_a(x)|y(X) = F] \\
&= m_p(x) + m_a(x)
\end{aligned}
\tag{12}
$$

so $m_p$ and $m_a$ can be interpreted as the decomposition of $m$ into it's periodic and aperiodic components. However, there is no similar decomposition of the variance: $v(x) \neq v_p(x) + v_a(x)$ since $y_p$ and $y_a$ are not independent given the observations.

The sub-models can be interpreted as usual Gaussian process models with correlated noise. For example, $m_p$ is the best predictor based on kernel $k_p$ with an observational noise given by $k_a$. For a detailed discussion on the decomposition of models based on a sum of kernels, see *Durrande, Ginsbourger & Roustant (2012)*.

We now illustrate this model decomposition on the test function $f(x) = \sin(x) + x/20$ defined over $[0, 20]$. Figure 3 shows the obtained model after estimating $(\sigma_p^2, \ell_p, \sigma_a^2, \ell_a)$ of a decomposed Matérn 5/2 kernel. In this example, the estimated values of the lengthscales are very different allowing the model to capture efficiently the periodic component of the signal and the low frequency trend.

## APPLICATION TO MATÉRN KERNELS

The Matérn class of kernels provides a flexible class of stationary covariance functions for a Gaussian process model. The family includes the infinitely smooth exponentiated quadratic (i.e., Gaussian or squared exponential or radial basis function) kernel as well

as the non-differentiable Ornstein–Uhlenbeck covariance. In this section, we show how the Matérn class of covariance functions can be decomposed into periodic and aperiodic subspaces in the RKHS.

Matérn kernels $k$ are stationary kernels, which means that they only depend on the distance between the points at which they are evaluated: $k(x,y) = \tilde{k}(|x-y|)$. They are often introduced by the spectral density of $\tilde{k}$ (*Stein, 1999*):

$$S(\omega) = \left( \frac{\Gamma(\nu)\ell^{2\nu}}{2\sigma^2\sqrt{\pi}\,\Gamma(\nu+1/2)(2\nu)^{\nu}} \left( \frac{2\nu}{\ell^2} + \omega^2 \right)^{\nu+1/2} \right)^{-1}. \tag{13}$$

Three parameters can be found in this equation: $\nu$ which tunes the differentiability of $\tilde{k}$, $\ell$ which corresponds to a lengthscale parameter and $\sigma^2$ that is homogeneous to a variance.

The actual expressions of Matérn kernels are simple when the parameter $\nu$ is half-integer. For $\nu = 1/2, 3/2, 5/2$ we have

$$k_{1/2}(x,x') = \sigma^2 \exp\left( -\frac{|x-x'|}{\ell} \right)$$

$$k_{3/2}(x,x') = \sigma^2 \left( 1 + \frac{\sqrt{3}|x-x'|}{\ell} \right) \exp\left( -\frac{\sqrt{3}|x-x'|}{\ell} \right) \tag{14}$$

$$k_{5/2}(x,x') = \sigma^2 \left( 1 + \frac{\sqrt{5}|x-x'|}{\ell} + \frac{5|x-x'|^2}{3\ell^2} \right) \exp\left( -\frac{\sqrt{5}|x-x'|}{\ell} \right).$$

Here the parameters $\ell$ and $\sigma^2$ respectively correspond to a rescaling of the abscissa and ordinate axis. For $\nu = 1/2$ one can recognise the expression of the exponential kernel (i.e., the covariance of the Ornstein–Uhlenbeck process) and the limit case $\nu \to \infty$ corresponds to the squared exponential covariance function (*Rasmussen & Williams, 2006*).

As stated in *Porcu & Stein* (*2012* Theorem 9.1) and *Wendland (2005)*, the RKHS generated by $k_\nu$ coincides with the Sobolev space $W_2^{\nu+1/2}$. Since the elements of the Fourier basis are $\mathcal{C}^\infty$, they belong to the Sobolev space and thus to Matérn RKHS. The hypothesis $\mathcal{H}_p \subset \mathcal{H}$ made in 'Kernels of Periodic and Aperiodic Subspaces' is thus fulfilled and all previous results apply.

Furthermore, the connection between Matérn kernels and autoregressive processes allows us to derive the expression of the RKHS inner product. As detailed in Appendix B, we obtain for an input space $D = [a,b]$:

Matérn 1/2 (exponential kernel)

$$\langle g, h \rangle_{\mathcal{H}_{1/2}} = \frac{\ell}{2\sigma^2} \int_a^b \left( \frac{1}{\ell}g + g' \right) \left( \frac{1}{\ell}h + h' \right) dt + \frac{1}{\sigma^2} g(a)h(a). \tag{15}$$

Matérn 3/2

$$\langle g, h \rangle_{\mathcal{H}_{3/2}} = \frac{\ell^3}{12\sqrt{3}\sigma^2} \int_a^b \left( \frac{3}{\ell^2}g + 2\frac{\sqrt{3}}{\ell}g' + g'' \right) \left( \frac{3}{\ell^2}h + 2\frac{\sqrt{3}}{\ell}h' + h'' \right) dt$$

$$+ \frac{1}{\sigma^2}g(a)h(a) + \frac{\ell^2}{3\sigma^2}g'(a)h'(a). \tag{16}$$

Matérn 5/2

$$\langle g,h\rangle_{\mathcal{H}_{5/2}} = \int_a^b L_t(g)L_t(h)\mathrm{d}t + \frac{9}{8\sigma^2}g(a)h(a) + \frac{9\ell^4}{200\sigma^2}g(a)''h''(a)$$
$$+ \frac{3\ell^2}{5\sigma^2}\left(g'(a)h'(a) + \frac{1}{8}g''(a)h(a) + \frac{1}{8}g(a)h''(a)\right) \tag{17}$$

where

$$L_t(g) = \sqrt{\frac{3\ell^5}{400\sqrt{5}\sigma^2}}\left(\frac{5\sqrt{5}}{\ell^3}g(t) + \frac{15}{\ell^2}g'(t) + \frac{3\sqrt{5}}{\ell}g''(t) + g'''(t)\right).$$

Although these expressions are direct consequences of *Doob (1953)* and *Hájek (1962)*, they cannot be found in the literature to the best of our knowledge.

The knowledge of these inner products allow us to compute the Gram matrix $G$ and thus the sub-kernels $k_p$ and $k_a$. A result of great practical interest is that inner products between the basis functions have a closed form expression. Indeed, all the elements of the basis can be written in the form $\cos(\omega x + \varphi)$ and, using the notation $L_x$ for the linear operators in the inner product integrals (see Eq. (17)), we obtain:

$$L_x(\cos(\omega x + \varphi)) = \sum_i \alpha_i \cos(\omega x + \varphi)^{(i)} = \sum_i \alpha_i \omega^i \cos\left(\omega x + \varphi + \frac{i\pi}{2}\right). \tag{18}$$

The latter can be factorised in a single cosine $\rho\cos(\omega x + \phi)$ with

$$\rho = \sqrt{r_c^2 + r_s^2}, \qquad \phi = \begin{cases} \arcsin(r_s/\rho) & \text{if } r_c \geq 0 \\ \arcsin(r_s/\rho) + \pi & \text{if } r_c < 0 \end{cases} \tag{19}$$

where

$$r_c = \sum_i \alpha_i \omega^i \cos\left(\varphi + \frac{i\pi}{2}\right) \quad \text{and} \quad r_s = \sum_i \alpha_i \omega^i \sin\left(\varphi + \frac{i\pi}{2}\right).$$

Eventually, the computation of the inner product between functions of the basis boils down to the integration of a product of two cosines, which can be solved by linearisation.

## QUANTIFYING THE PERIODICITY

The decomposition of the model into a sum of sub-models is useful for quantifying the periodicity of the pseudo-periodic signals. In this section, we propose a criterion based on the ratio of signal variance explained by the sub-models.

In sensitivity analysis, a common approach for measuring the effect of a set of variables $x_1,\ldots,x_n$ on the output of a multivariate function $f(x_1,\ldots,x_n)$ is to introduce a random vector $R = (r_1,\ldots,r_n)$ with values in the input space of $f$ and to define the variance explained by a subset of variables $x_I = (x_{I_1},\ldots,x_{I_m})$ as $V_I = \mathrm{Var}\big(\mathrm{E}\big(f(R)|R_I\big)\big)$ (*Oakley & O'Hagan, 2004*). Furthermore, the prediction variance of the Gaussian process model can be taken into account by computing the indices based on random paths of the conditional Gaussian process (*Marrel et al., 2009*).

We now apply these two principles to define a periodicity ratio based on the sub-models. Let $R$ be a random variable defined over the input space and $y_p$, $y_a$ be the periodic and aperiodic components of the conditional process $y$ given the data-points. $y_p$ and $y_a$ are normally distributed with respective mean and variance $(m_p, v_p)$, $(m_a, v_a)$ and their covariance is given by $\mathrm{Cov}(y_p(x), y_a(x')) = -k_p(x, X)k(X, X)^{-1}k_a(x')$. To quantify the periodicity of the signal we introduce the following periodicity ratio:

$$S = \frac{\mathrm{Var}_R[y_p(R)]}{\mathrm{Var}_R[y_p(R) + y_a(R)]}. \tag{20}$$

Note that $S$ cannot be interpreted as a the percentage of periodicity of the signal in a rigorous way since $\mathrm{Var}_R[y_p(R) + y_a(R)] \neq \mathrm{Var}_R[y_p(R)] + \mathrm{Var}_R[y_a(R)]$. As a consequence, this ratio can be greater than 1.

For the model shown in Fig. 3, the mean and standard deviation of $S$ are respectively 0.86 and 0.01.

## APPLICATION TO GENE EXPRESSION ANALYSIS

The 24 h cycle of days can be observed in the oscillations of biological mechanisms at many spatial scales. This phenomenon, called the circadian rhythm, can for example be seen at a microscopic level on gene expression changes within cells and tissues. The cellular mechanism ensuring this periodic behaviour is called the circadian clock. For arabidopsis, which is a widely used organism in plant biology and genetics, the study of the circadian clock at a gene level shows an auto-regulatory system involving several genes (*Ding et al., 2007*). As argued by *Edwards et al. (2006)*, it is believed that the genes involved in the oscillatory mechanism have a cyclic expression so the detection of periodically expressed genes is of great interest for completing current models.

Within each cell, protein-coding genes are transcribed into messenger RNA molecules which are used for protein synthesis. To quantify the expression of a specific protein-coding gene it is possible to measure the concentration of messenger RNA molecules associated with this gene. Microarray analysis and RNA-sequencing are two examples of methods that take advantage of this principle.

The dataset (see http://millar.bio.ed.ac.uk/data.htm) considered here was originally studied by *Edwards et al. (2006)*. It corresponds to gene expression for nine day old arabidopsis seedlings. After eight days under a 12 h-light/12 h-dark cycle, the seedlings are transferred into constant light. A microarray analysis is performed every four hours, from 26 to 74 h after the last dark-light transition, to monitor the expression of 22,810 genes. *Edwards et al. (2006)* use COSOPT (*Straume, 2004*) for detecting periodic genes and identify a subset of 3,504 periodically expressed genes, with an estimated period between 20 and 28 h.

We now apply to this dataset the method described in the previous sections. The kernel we consider is a sum of a periodic and aperiodic Matérn 3/2 kernel plus a delta function to reflect observation noise:

$$k(x, x') = \sigma_p^2 k_p(x, x') + \sigma_a^2 k_a(x, x') + \tau^2 \delta(x, x'). \tag{21}$$

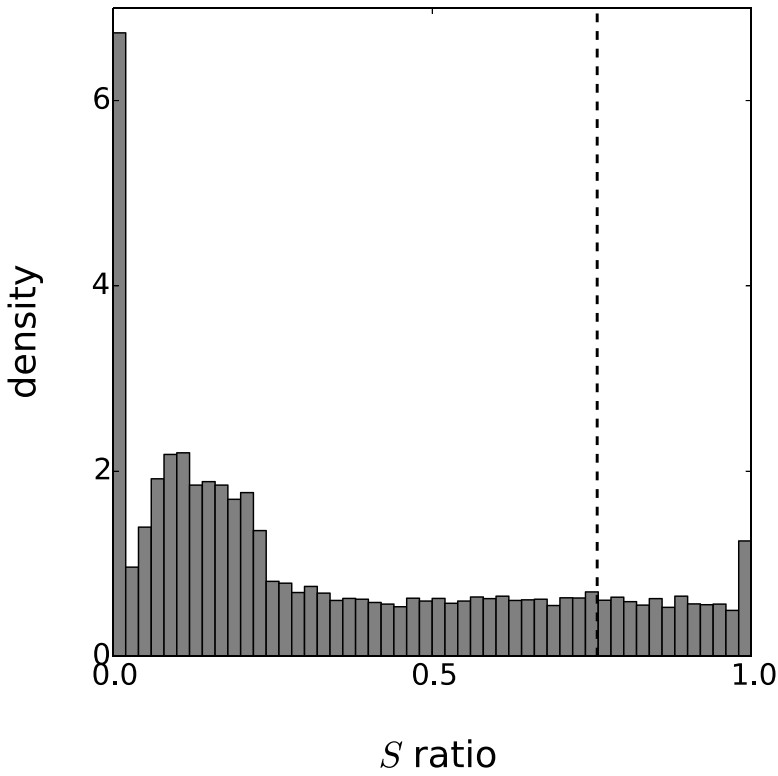

**Figure 4** **Distribution of the periodicity ratio over all genes according to the Gaussian process models.** The cut-off ratio determining if genes are labelled as periodic or not is represented by a vertical dashed line.

Although the cycle of the circadian clock is known to be around 24 h, circadian rhythms often depart from this figure (indeed *circa dia* is Latin for *around a day*) so we estimated the parameter $\lambda$ to determine the actual period. The final parametrisation of $k$ is based on six variables: $(\sigma_p^2, \ell_p, \sigma_a^2, \ell_a, \tau^2, \lambda)$. For each gene, the values of these parameters are estimated using maximum likelihood. The optimization is based on the standard options of the GPy toolkit with the following boundary limits for the parameters: $\sigma_p, \sigma_a \geq 0$; $\ell_p, \ell_a \in [10, 60]$; $\tau^2 \in [10^{-5}, 0.75]$ and $\lambda \in [20, 28]$. Furthermore, 50 random restarts are performed for each optimization to limit the effects of local minima.

Eventually, the periodicity of each model is assessed with the ratio $S$ given by Eq. (20). As this ratio is a random variable, we approximate the expectation of $S$ with the mean value of 1,000 realisations. To obtain results comparable with the original paper on this dataset, we labeled as periodic the set of 3,504 genes with the highest periodicity ratio. The cut-off periodicity ratio associated with this quantile is $S = 0.76$. As can be seen in Fig. 4, this cut-off value does not appear to be of particular significance according to the distribution of the Gaussian process models. On the other hand, the distribution spike that can be seen at $S = 1$ corresponds to a gap between models that are fully-periodic and others. We believe this gap is due to the maximum likelihood estimation since the estimate of $\sigma_a^2$ is zero for all models in the bin $S = 1$. The other spike at $S = 0$ can be interpreted similarly and it corresponds to estimated $\sigma_p^2$ equal to zero.

**Table 1** Confusion table associated to the predictions by COSOPT and the proposed Gaussian process approach.

| # of genes | $\mathcal{P}_{GP}$ | $\overline{\mathcal{P}_{GP}}$ |
|---|---|---|
| $\mathcal{P}_{COSOPT}$ | 2,127 | 1,377 |
| $\overline{\mathcal{P}_{COSOPT}}$ | 1,377 | 17,929 |

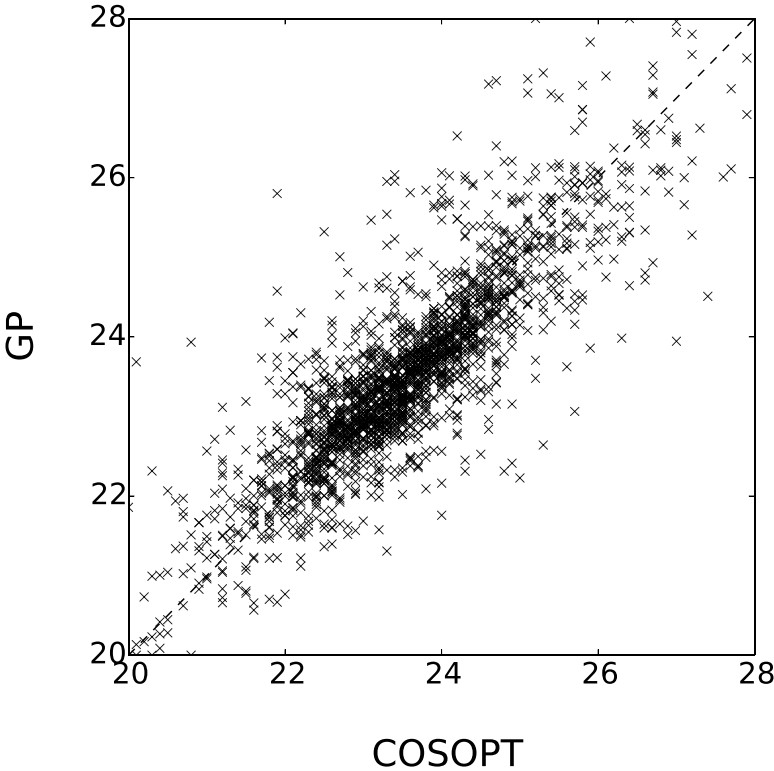

**Figure 5** Comparison of Estimated periods for the genes in $\mathcal{P}_{GP} \cap \mathcal{P}_{COSOPT}$. The coefficient of determination of $x \rightarrow x$ (dashed line) is 0.69.

Let $\mathcal{P}_{COSOPT}$ and $\mathcal{P}_{GP}$ be the sets of selected periodic genes respectively by *Edwards et al. (2006)* and the method presented here and let $\overline{\mathcal{P}_{COSOPT}}$ and $\overline{\mathcal{P}_{GP}}$ denote their complements. The overlap between these sets is summarised in Table 1. Although the results cannot be compared to any ground truth, the methods seem coherent since 88% of the genes share the same label. Furthermore, the estimated value of the period λ is consistent for the genes labelled as periodic by the two methods, as seen in Fig. 5.

One interesting comparison between the two methods is to examine the genes that are classified differently. The available data from *Edwards et al. (2006)* allows focusing on the worst classification mistakes made by one method according to the other. This is illustrated in Fig. 6 which shows the behaviour of the most periodically expressed genes in $\overline{\mathcal{P}_{GP}}$ according to COSOPT and, conversely, the genes in $\overline{\mathcal{P}_{COSOPT}}$ with the highest periodicity ratio $S$. Although it is undeniable that the genes selected only by COSOPT (Fig. 6A) present some periodic component, they also show a strong non-periodic part,

PeerJ Computer Science

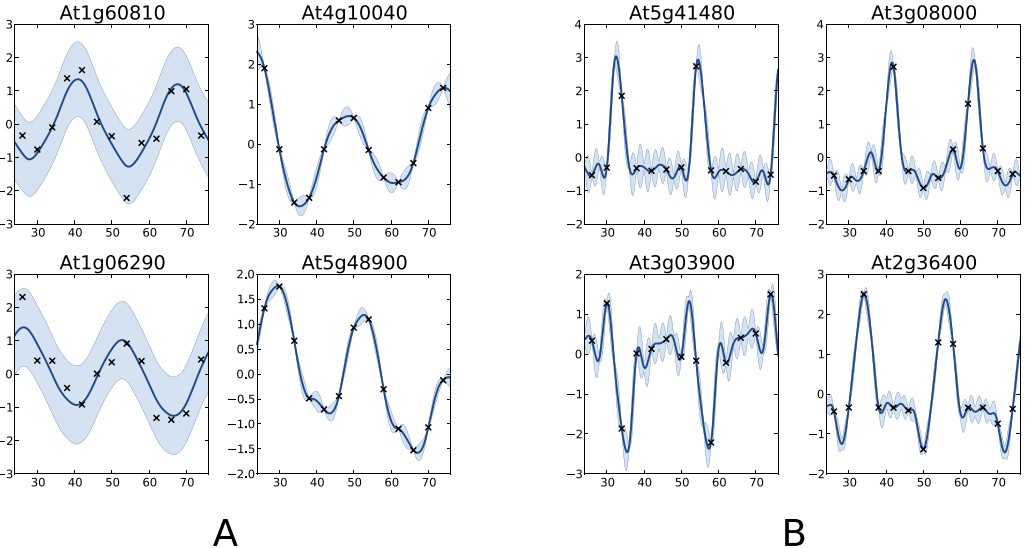

**Figure 6** **Examples of genes with different labels.** (A) corresponds to genes labelled as periodic by COSOPT but not by the Gaussian process approach, whereas in (B) they are labelled as periodic only by the latter. In (A, B), the four selected genes are those with the highest periodic part according to the method that labels them as periodic. The titles of the graphs correspond to the name of the genes (AGI convention).

corresponding either to noise or trend. For these genes, the value of the periodicity ratio is: 0.74 (0.10), 0.74 (0.15), 0.63 (0.11), 0.67 (0.05) (means and standard deviations, clockwise from top left) which is close to the classification boundary. On the other hand, the genes selected only by the Gaussian process approach show a strong periodic signal (we have for all genes $S = 1.01$ (0.01)) with sharp spikes. We note from Fig. 6B that there is always at least one observation associated with each spike, which ensures that the behaviour of the Gaussian process models cannot simply be interpreted as overfitting. The reason COSOPT is not able to identify these signals as periodic is that it is based on a single cosine function which makes it inadequate for fitting non sinusoidal periodic functions. This is typically the case for gene expressions with spikes as in Fig. 6B, but it can also be seen on the test functions of Fig. 1.

This comparison shows very promising results, both for the capability of the proposed method to handle large datasets and for the quality of the results. Furthermore, we believe that the spike shape of the newly discovered genes may be of particular interest for understanding the mechanism of the circadian clock. The full results, as well as the original dataset, can be found in the Supplemental Information.

## CONCLUSION

The main purpose of this article is to introduce a new approach for estimating, extracting and quantifying the periodic component of a pseudo-periodic function $f$ given some noisy observations $y_i = f(x_i) + \varepsilon$. The proposed method is typical in that it corresponds to the orthogonal projection onto a basis of periodic functions. The originality here is to perform this projection in some RKHS where the partial knowledge given by the observations can

Durrande et al. (2016), *PeerJ Comput. Sci.*, DOI 10.7717/peerj-cs.50                    **13/18**

be dealt with elegantly. Previous theoretical results from the mid-1900s allowed us to derive the expressions of the inner product of RKHS based on Matérn kernels. Given these results, it was then possible to define a periodic kernel $k_p$ and to decompose $k$ as a sum of sub-kernels $k = k_p + k_a$.

We illustrated three fundamental features of the proposed kernels for Gaussian process modelling. First, as we have seen on the benchmark examples, they allowed us to approximate periodic non-sinusoidal patterns while retaining appropriate filtering of the noise. Second, they provided a natural decomposition of the Gaussian process model as a sum of periodic and aperiodic sub-models. Third, they can be reparametrised to define a wider family of kernel which is of particular interest for decoupling the assumptions on the behaviour of the periodic and aperiodic part of the signal.

The probabilistic interpretation of the decomposition in sub-models is of great importance when it comes to define a criterion that quantifies the periodicity of $f$ while taking into account the uncertainty about it. This goal was achieved by applying methods commonly used in Gaussian process based sensitivity analysis to define a periodicity ratio.

Although the proposed method can be applied to any time series data, this work has originally been motivated by the detection of periodically expressed genes. In practice, listing such genes is a key step for a better understanding of the circadian clock mechanism at the gene level. The effectiveness of the method is illustrated on such data in the last section. The results we obtained are consistent with the literature but they also feature some new genes with a strong periodic component. This suggests that the approach described here is not only theoretically elegant but also efficient in practice.

As a final remark, we would like to stress that the proposed method is fully compatible with all the features of Gaussian processes, from the combination of one-dimensional periodic kernels to obtain periodic kernels in higher dimension to the use of sparse methods when the number of observation becomes large. By implementing our new method within the GPy package for Gaussian process inference we have access to these generalisations along with effective methods for parameter estimation. An interesting future direction would be to incorporate the proposed kernel into the 'Automated Statistician' project (*Lloyd et al., 2014*; *Duvenaud et al., 2013*), which searches over grammars of kernels.

## APPENDIX A. DETAILS ON TEST FUNCTIONS

The test functions shown in Fig. 1 are 1-periodic. Their expressions for $x \in [0, 1)$ are (from top left, in a clockwise order):

$$f_1(x) = \cos(2\pi x)$$
$$f_2(x) = 1/2\cos(2\pi x) + 1/2\cos(4\pi x)$$
$$f_3(x) = \begin{cases} 1 & \text{if } x \in [0, 0.2] \\ -1 & \text{if } x \in (0.2, 1) \end{cases} \quad (22)$$
$$f_4(x) = 4|x - 0.5| + 1$$
$$f_5(x) = 1 - 2x$$
$$f_6(x) = 0.$$

# APPENDIX B. NORMS IN MATÉRN RKHS

## Autoregressive processes and RKHS norms

A process is said to be autoregressive (AR) if the spectral density of the kernel

$$S(\omega) = \frac{1}{2\pi} \int_{\mathbb{R}} k(t) e^{-i\omega t} \, \mathrm{d}t \tag{23}$$

can be written as a function of the form

$$S(\omega) = \frac{1}{\left| \sum_{k=0}^{m} \alpha_k (i\omega)^k \right|^2} \tag{24}$$

where the polynomial $\sum_{k=0}^{m} \alpha_k x^k$ is real with no zeros in the right half of the complex plan *Doob (1953)*. Hereafter we assume that $m \geq 1$ and that $\alpha_0, \alpha_m \neq 0$.

For such kernels, the inner product of the associated RKHS $\mathcal{H}$ is given by *Hájek (1962)*, *Kailath (1971)* and *Parzen (1961)*

$$\langle h, g \rangle_{\mathcal{H}} = \int_a^b (L_t h)(L_t g) \mathrm{d}t + 2 \sum_{\substack{0 \leq j, k \leq m-1 \\ j+k \text{ even}}} d_{j,k} h^{(j)}(a) g^{(k)}(a) \tag{25}$$

where

$$L_t h = \sum_{k=0}^{m} \alpha_k h^{(k)}(t) \quad \text{and} \quad d_{j,k} = \sum_{i=\max(0, j+k+1-n)}^{\min(j,k)} (-1)^{(j-i)} \alpha_i \alpha_{j+k+1-i}.$$

We show in the next section that the Matérn kernels correspond to autoregressive kernels and, for the usual values of $\nu$, we derive the norm of the associated RKHS.

## Application to Matérn kernels

Following the pattern exposed in *Doob* (*1953*, p. 542), the spectral density of a Matérn kernel (Eq. (13)) can be written as the density of an AR process when $\nu + 1/2$ is an integer. Indeed, the roots of the polynomial $\frac{2\nu}{\ell^2} + \omega^2$ are conjugate pairs so it can be expressed as the squared module of a complex number

$$\frac{2\nu}{\ell^2} + \omega^2 = \left( \omega + \frac{i\sqrt{2\nu}}{\ell} \right) \left( \omega - \frac{i\sqrt{2\nu}}{\ell} \right) = \left| \omega + \frac{i\sqrt{2\nu}}{\ell} \right|^2. \tag{26}$$

Multiplying by $i$ and taking the conjugate of the quantity inside the module, we finally obtain a polynomial in $i\omega$ with all roots in the left half of the complex plan:

$$\frac{2\nu}{\ell^2} + \omega^2 = \left| i\omega + \frac{\sqrt{2\nu}}{\ell} \right|^2 \Rightarrow \left( \frac{2\nu}{\ell^2} + \omega^2 \right)^{(\nu+1/2)} = \left| \left( \frac{\sqrt{2\nu}}{\ell} + i\omega \right)^{(\nu+1/2)} \right|^2. \tag{27}$$

Plugging this expression into Eq. (13), we obtain the desired expression of $S_\nu$:

$$S_\nu(\omega) = \frac{1}{\left| \sqrt{\frac{\Gamma(\nu)\ell^{2\nu}}{2\sigma^2 \sqrt{\pi} \Gamma(\nu+1/2)(2\nu)^\nu}} \left( \frac{\sqrt{2\nu}}{\ell} + i\omega \right)^{(\nu+1/2)} \right|^2}. \tag{28}$$

Using $\Gamma(\nu) = \frac{(2\nu-1)!\sqrt{\pi}}{2^{2\nu-1}(\nu-1/2)!}$, one can derive the following expression of the coefficients $\alpha_k$:

$$\alpha_k = \sqrt{\frac{(2\nu-1)!\nu^\nu}{\sigma^2(\nu-1/2)!^2 2^\nu}} C_{\nu+1/2}^k \left(\frac{\ell}{\sqrt{2\nu}}\right)^{k-1/2}. \tag{29}$$

Theses values of $\alpha_k$ can be plugged into Eq. (25) to obtain the expression of the RKHS inner product. The results for $\nu \in \{1/2, 3/2, 5/2\}$ is given by Eqs. (15)–(17) in the main body of the article.

### Funding
Support was provided by the BioPreDynProject (Knowledge Based Bio-Economy EU grant Ref 289434) and the BBSRC grant BB/1004769/1. James Hensman was funded by an MRC career development fellowship. The funders had no role in study design, data collection and analysis, decision to publish, or preparation of the manuscript.

### Grant Disclosures
The following grant information was disclosed by the authors:
BioPreDynProject: Ref 289434.
BBSRC: BB/1004769/1.

### Competing Interests
The authors declare there are no competing interests.

### Author Contributions
- Nicolas Durrande and James Hensman conceived and designed the experiments, performed the experiments, analyzed the data, contributed reagents/materials/analysis tools, wrote the paper, prepared figures and/or tables, performed the computation work, reviewed drafts of the paper.
- Magnus Rattray and Neil D. Lawrence conceived and designed the experiments, analyzed the data, contributed reagents/materials/analysis tools, wrote the paper, performed the computation work, reviewed drafts of the paper.

### Data Availability
GitHub: https://github.com/SheffieldML.

### Supplemental Information
Supplemental information for this article can be found online at http://dx.doi.org/10.7717/peerj-cs.50#supplemental-information.

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
