# Peer review of "Detecting periodicities with Gaussian processes"

_PeerJ Computer Science, doi:10.7717/peerj-cs.50_

## Round 0.1 · original submission · Minor Revisions

· Academic Editor

Minor Revisions

The reviewer consensus is that the paper is well written and will provide new tools for detecting periodicity in large data sets. The motivating example is relevant to cellular biologists and will be interesting to a broader class of readers. The paper is ready for publication once the following minor issues have been addressed.

Reviewer 1 has identified some typographical errors that need to be fixed. Reviewer 1 also asks that you discuss the sensitivity of your results to the 0.77 threshold.

In addition, it would be good to include a little more discussion of the right-hand panel of Figure 5. The authors note that the disagreements on the left-hand side are near-misses for GP. But no real insight is given into why the methods disagree on the cases in the right-hand panel. Why didn't CSOPT find periodicity there? The paper says the comparison allows us to focus on the differences between the methods, and that some new genes with a strong periodic component have been identified. This is potentially very interesting biologically. But there is no analysis or explanation of why those periodicities were not found by other methods.

Finally, there are some additional problems that need to be corrected beyond those found by Reviewer 1:

1. Page 4: "a process y_p which samples are periodic functions" - This is ungrammatical. Do you mean "the samples of which are periodic functions"?

2. Page 4: matrix notations --> matrix notation

3. Page 7: I have never seen the term "degenerated parameters." What is that? Do you mean parameter settings for which H_p or H_a is a degenerate Hilbert space? A degenerate Hilbert space is a well-defined notion; degenerated parameters is non-standard. If this is really what you mean and the term has a precise meaning, you need to define it.

4. Page 7: Equation (10) is introduced as a decomposition of the best predictor as a sum of sub-models m_a and m_p. Then you need to say what m_a and m_p are. Remind the reader that a Chaussian process model is characterized through its mean and variance. State explicitly that E[y_p(x)|y(X)=F] and E[y_a(x)|y(X)=F] are the means of the two sub-models. You do this in Equation (11) for the variances, but you never explicitly define An alert reader familiar with Gaussian processes can figure out that this is what you mean, but as written it could be confusing for the uninitiated.

5. independent knowing the observations --> independent given the observations (or independent conditional on the observations) The terminology "knowing" for conditioning also appears elsewhere, and is non-standard.

6. Page 8: points they are evaluated at --> points at which they are evaluated

7. Page 9: they belongs to --> they belong to

8. Page 10: knowing the data-points --> given the data points

9. Page 11: As advocated by Edwards et al. --> As hypothesized by Edwards et al. (or argued, or studied, or discussed) Science proceeds by hypothesizing and evaluating empirically. Advocacy is for lobbyists.

10. Page 12: cannot be compare to --> cannot be compared to

11. Page 14: There is a space before the period at the end of the second-to-last sentence in the first paragraph.

11. As often, the proposed method ... --> The proposed method is typical in that it... "As often" is not standard English.

Reviewer 1 ·

Basic reporting

This is a well-written manuscript, with very little to criticise. It is nicely self-contained, and focused. The use of iPython/jupyter notebooks is to be commended, as well as the availability of code and data.

Current references to relevant literature seem appropriate. I did wonder if it might be worth making reference to the "automatic statistician" project (of Ghahramani and others). However, I am certain the authors are aware of this project, and am happy to leave this to their discretion. I was also a little surprised that the authors did not cite more of their own relevant work.

I spotted just a few typos/grammatical errors, listed below:

1. Page 5: "which samples are periodic functions" and "This can obtain by".

2. Page 13: "Although the results cannot be compare to any ground truth"

3. Page 15: "We illustrated three fundamental feature of"

4. Consistency with "arabidopsis" vs. "Arabidopsis" (and italics). Perhaps not so important for a computer science audience, but it is nevertheless nice to be consistent. [Since arabidopsis has become the common name for A. thaliana, I personally think neither capitalisation nor italics are required if the name is just given as arabidopsis -- which looks to be the case throughout this manuscript].

Experimental design

The experimental design appears sound, with the work conducted rigorously and to a high technical standard. Reproducibility is ensured through open-source implementation, and the use of iPython/jupyter notebooks.

I am certainly not an expert on reproducing kernel Hilbert spaces, but I did not spot any obvious blunders.

Validity of the findings

The application of the proposed method to gene expression data is concise and to the point. The comparison to COSOPT is welcomed, and the level of agreement between these methods is reassuring.

I have just one query, which is minor but seems potentially important for future applications of the method. As far as I understood, the classification threshold for the A. thaliana dataset was determined to ensure that the number of periodic genes was the same as the number returned by COSOPT (i.e. 3504). But, in practice, how would I determine an "optimal" classification threshold for the proposed method, without running COSOPT? For this particular example, could it be the case that the "optimal" classification threshold is quite a lot lower than 0.77, and actually all of the periodic genes reported by COSOPT would be reported by the proposed method (if this lower threshold had been chosen)? That is, could it be the case that the effective difference between the methods (in practice) is one of having different sensitivity/specificity properties?

·

Basic reporting

This is a self contained paper describing how additive covariance functions for Gaussian Process priors can be used to detect systematic periodicity in observed data. The motivating example of circadian enzymatic control is highly relevant to contemporary cellular biology and a good example of where this sort of statistical machinery proves to be useful.

Experimental design

This is all appropriate for the study undertaken.

Validity of the findings

In terms of the statistical methodology there is sufficient validation of the outcomes.

Additional comments

A reasonably comprehensive and self-contained article that will introduce readers to the Gaussian Process machinery of inference and how covariance functions may be constructed to address a specific study, in this case detecting periodicities. Well written and clear to read.

---

## Round 0.2 · accepted · Accept

· Academic Editor

Accept

The revised version of this paper satisfactorily addresses the issues raised by the reviewers and the additional issues I raised in my decision letter. The paper as currently written is suitable for publication.